# Evaluation of Biogeochemical Changes in Channelized and Restored Portions of a Subtropical Floodplain

**Paul Julian II [1,2,\*], Todd Z. Osborne [2,3] and Rex Ellis [3,4]**

1    Sanibel-Captiva Conservation Foundation, Sanibel, FL 33957, USA
2    Whitney Laboratory for Marine Bioscience, University of Florida, St. Augustine, FL 32080, USA
3    Department of Soil, Water and Ecosystem Sciences, University of Florida, Gainesville, FL 32611, USA
4    St. Johns River Water Management District, Palatka, FL 32177, USA
\*    Correspondence: pjulian@sccf.org

**Abstract:** Floodplains are critically important ecosystems that provide a whole suite of ecosystem services, including nutrient and carbon sequestration, flood mitigation, water storage, and critical wildlife habitat. However, human modification of rivers and floodplains through channelization, artificial levee construction, reductions in the active floodplain area, and water management can significantly reduce the ecosystem function of river–floodplain systems. In this study, we evaluated the changes in the nutrient loading of the Kissimmee River floodplain during the restoration of the river–floodplain system. In addition to time-series loading analysis, we also evaluated soil nutrient concentrations across the lower portion of the Kissimmee River floodplain. During the 44-year nutrient loading time-series, the floodplain remained a nutrient exporter with changes in nutrient loading generally corresponding to both water quality (i.e., point source reductions) and hydrologic restoration activities in the watershed and Kissimmee River floodplain. During the study period, inputs of total phosphorus and total nitrogen loads from upstream either significantly increased or remained the same. In addition to external sources of nutrients, internal sources of nutrients from floodplain soils can also contribute to the total nutrient export from the system. These internal sources could be organic via the decomposition of organic matter or geologic from the original excavation of the canal and/or restoration backfilling. Soil nutrient concentrations vary between vegetative communities and landscape position and could be a significant source of phosphorus to the downstream system, which is plagued by eutrophic conditions. Therefore, as floodplain function in the Kissimmee River continues to be restored and managed, additional effort may be needed to address nutrient inputs and internal legacy nutrients.

**Keywords:** floodplain restoration; phosphorus; nutrients; loading





## 1. Introduction

Floodplains are characteristically dynamic environments located at the aquatic–terrestrial interface governed by climate and flow regimes and balances between lotic and lentic conditions. Floodplains are not homogeneous ecological units but rather a mosaic of distinct hydrologic environments [1]. While the definition of floodplains can vary, the genetic floodplain definition includes hydraulic, hydrologic, and geomorphic features. Nanson and Croke (1992) define the genetic floodplain as "the largely horizontally-bedded alluvial landform adjacent to a channel, separated from the channel by banks and built of sediment transported by the present flow-regime" [2]. Therefore, floodplains are defined in terms of hydrology, such as inundation frequency, and geomorphic terms concerning inundation and depositional history.

River–floodplain systems provide critical habitat for terrestrial and aquatic biota as well as key ecosystem services, including water storage, flood mitigation, carbon sequestration, and nutrient retention [1–3]. Floodplains perform a diverse array of interconnected

physical and ecological functions [1,3,4]. Primarily floodplains attenuate fluxes of water, nutrients, and particulate material such as sediment and organic matter. How these constituents are attenuated depends on the river and floodplain morphology, which in turn dictate how things move downstream [4–6]. Therefore, changes to floodplain extent, surface water inputs, and climate variables can affect floodplain function.

Human alterations along stream channels, rivers, and within catchments have affected floodplain ecological functions by changing fluvial geomorphic processes, floodplain morphology, and channel–floodplain connectivity. Human alterations include changes to water management; water control feature construction, such as dams, levees, canals, and channelization; land clearance with upland erosion; and downstream aggradation that can lead to geomorphic changes to a floodplain and the disconnection of channels and floodplains, affecting vertical accretion and storage [1,7,8]. These changes cause a cascade of impacts on the natural ecology of floodplains by altering suitable habitats, biodiversity, and nutrient cycling [7,9]. Artificial levees, channelization, and water management can cause complex changes in river–floodplain connectivity, reducing floodplain extent and fundamentally altering the habitat of aquatic and wetland communities [6,10].

A prime example of anthropogenic manipulation of river–floodplain ecosystems (and its restoration) is the Kissimmee River. Historically, the Kissimmee River meandered 161 km from Lake Kissimmee through a 2–5 km wide floodplain eventually flowing into Lake Okeechobee. The river was characterized by a low elevation gradient (0.07 m km$^{-1}$) and a mean main channel velocity of 0.2–0.6 m s$^{-1}$ [11,12]. This hydrology and associated hydrodynamics shaped the flora and fauna of the Kissimmee River floodplain. As such, the floodplain wetlands were characterized by a diverse assemblage of plant communities adapted for long-duration inundation patterns and supported abundant wildlife [10,13]. Long-duration flooding was common in the pre-drainage Kissimmee Basin with little adverse impact on human life and property; however, after severe flooding, flood control measures were initiated in the mid to late 1940s in the form of the Central & Southern Flood Control Project. Construction of a flood control canal (C-38) was initiated in the lower Kissimmee Basin in 1962 and completed less than a decade later. The canal bisected the meandering Kissimmee River, eliminating flow in remnant channels and conveying all water formerly in the floodplain to Lake Okeechobee. A series of water control structures were also constructed along C-38 to control releases of water to and through the basin. The effect of channelization was realized as wetland ecosystems quickly deteriorated, relatively consistent water levels were managed throughout the different pools, and seasonal overbank flow was eliminated [12,14,15]. The Kissimmee River restoration project was recently completed (July 2021) and backfilled 35 km of canal, reconstructed ~18 km of river channel, opened up ~64 km of floodplain, and restored ~50 km$^2$ of wetlands. Ultimately, the river–floodplain physical template was reconstructed, and now changes to water management are needed to complete the hydrologic restoration of the floodplain [16].

An analysis of North American floodplains demonstrated the reduction in net connectivity within the Kissimmee River floodplain with >500 km$^2$ of accumulated alternating due to flood control measures with the same order of magnitude as other major floodplain systems in the U.S. Midwest and California [6]. This reduction in net connectivity ultimately impacts the ecological function of the river–floodplain system, including water storage and water quality functions. The primary goal of the Kissimmee River restoration effort was to restore the ecological integrity of the river–floodplain system by re-establishing the physical and hydrologic form of the river, including the seasonal changes in water management to the historic flood and drawdown dynamics. This restoration will thereby improve the net connectivity of the Kissimmee River floodplain and re-establish floodplain processes. While physical changes to the floodplain have occurred, first from the floodplain to the channelized system and then back again, changes to local and regional water quality have also occurred with accelerated eutrophication from different land-uses within the watershed [17,18]. The objectives of this study were to present a long-term study of nutrient loading on the Kissimmee River floodplain, document soil characteristics during a large-

scale restoration effort, and, by doing so, provide a baseline spatial distribution of these characteristics for use in future assessment of restoration efforts. Moreover, this is the first spatially intensive sampling of soil characteristics across the Kissimmee River floodplain. Our goal was to demonstrate the effect of incremental restoration on floodplain function relative to nutrient loading. We hypothesize that, as the restoration of the Kissimmee River floodplain progresses, nutrient export from the Kissimmee River will change.

## 2. Materials and Methods

### 2.1. Study Area

The Kissimmee River drains approximately 7804 km$^2$ of central Florida (Figure 1). The river begins as the outlet of Lake Kissimmee south of Orlando, Florida and ends at Lake Okeechobee. Historically, the river flowed for approximately 166 km in a low elevation gradient and branching channel that frequently flooded annually [10]. Historically, the river was a mosaic of wetland and upland vegetation communities with a broad marsh punctuated with live oak hummocks and a channel delineated by a narrow "levee" of willows and wetland shrubs [19]. Post channelization (1972), the channel was 9 m deep and approximately 100 m wide with the floodplain being separated into five pools created by size lock and dam structures constraining the floodplain into the C-38 canal. Incrementally over the past several decades, the Kissimmee River floodplain was restored through the removal of lock and dam structures (S65B 2001, S65C 2017; Table 1), backfilling of the C-38 canal, and degrading spoil areas (Table 1). The physical restoration of the Kissimmee River was recently completed (July 2021), paving the way to hydrologic and ecological restoration of the floodplain [16].

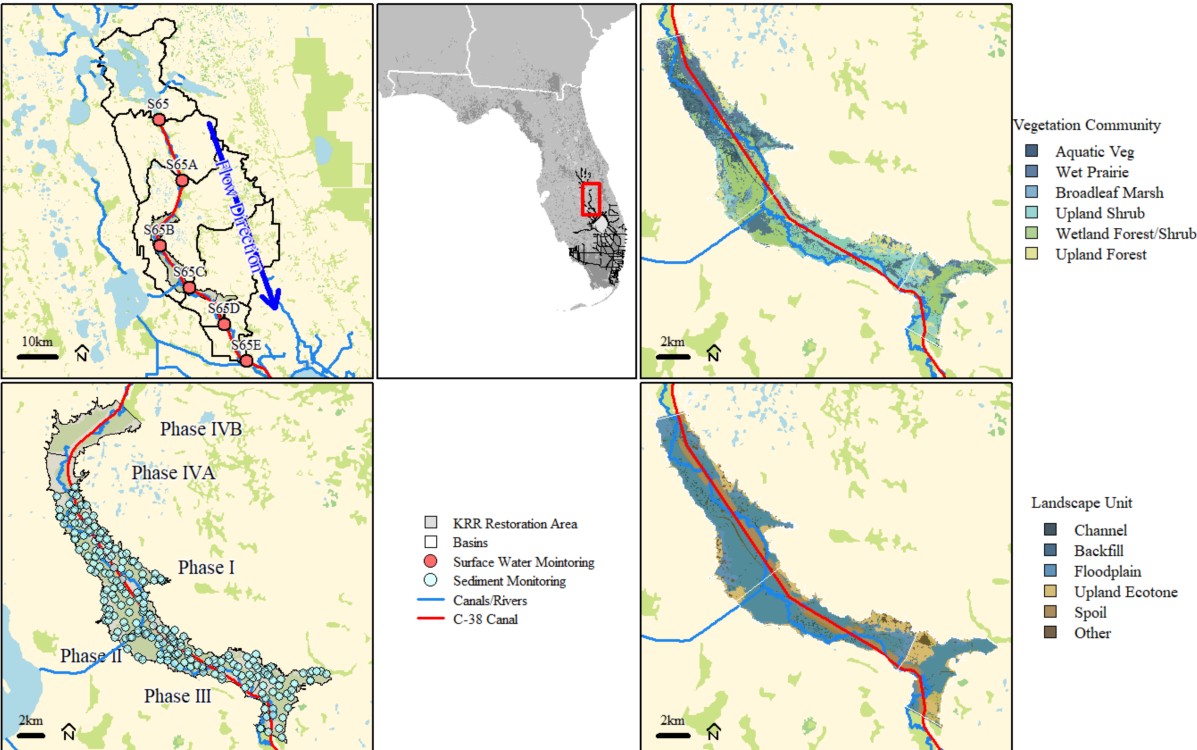

**Figure 1.** Surface water monitoring locations with the general direction of flow identified (**top left**) for the entire Kissimmee River basin; sediment monitoring locations (**bottom left**), overall study area (**top middle**), specific to the Phase I, II, and III segments of the Kissimmee River floodplain; vegetation (**top right**) and landscape unit (**bottom right**) coverages within the lower segments of the Kissimmee River floodplain.

**Table 1.** Major Actions and events related to the Kissimmee River Restoration Project and regional water control.

| Time Period | Major Action/Event |
| --- | --- |
| 1954 | United States Congress authorizes the C&SF Project |
| 1962 | Construction commences of the C38 Canal and other features |
| 1971 | Construction complete |
| 1971 | The state of Florida recommends the restoration of the Kissimmee River |
| 1978–1985 | Federal feasibility study notes the potential for restoration [20] |
| 1984–1990 | Kissimmee River Demonstration Project |
| 1987–1990 | Drawdown of upstream lakes (Lake Tohopekaliga and East Lake Tohopekaliga) [21] |
| 1991 | 2nd Federal feasibility study recommends a partial backfill plan [22] |
| 1994 | Construction of test backfill and performance of high-flow tests on backfill stability |
| 1995–1997 | Drawdown of upstream lakes (Lake Jackson and Lake Kissimmee) [21] |
| 1996 | Headwaters Revitalization Feasibility Study complete [23] |
| 1999–2001 | Phase I backfilling complete (12 km canal backfilled) |
| 2001 | S-65B water control structure removed |
| 2006–2010 | Phase IV backfilling complete (9 km canal backfilled) |
| 2010–2015 | Supplemental construction to support the Kissimmee River Restoration effort [24] |
| 2015–2021 | Phases II and III backfilling complete (~14 km canal backfilled) |
| 2017 | S-65C water control structure removed |
| 2021 | Kissimmee River Restoration Construction Activities complete |

### 2.2. Data Sources

Water quality and discharge data were retrieved from the South Florida Water Management District (SFWMD) online environmental database (DBHYDRO; www.sfwmd/dbhydro, accessed on 8 July 2022) for site routine monitoring locations along the Kissimmee River (Figure 1). Data collected between water year (WY) 1979 and 2022 (1 May 1979–30 April 2022) were considered for this study. This period was selected for several reasons: (1) it bookends the synoptic lake sediment surveys; (2) it includes several years where major hurricanes and tropical storms came in the vicinity of the lake; (3) it includes a period of a notable shift in the Atlantic Multidecadal Oscillation (AMO) index; and (4) it spans the period of pre-restoration and ongoing restoration activities within the greater Everglades ecosystem [16,25,26]. The water column parameters evaluated in this study include total phosphorus (TP) and TN. For the purposes of data analyses and summary statistics, data reported less than the method detection limit (MDL) were assigned a value of one-half the MDL.

Surface soil samples (0–10 cm) were collected at predetermined locations within Pools C and D (restoration Phases I and II/III; Figure 1) of the Kissimmee River floodplain. Samples were taken using two methods: a manual push corer and a piston corer. Manual push coring used stainless steel coring tubes (10 cm in diameter and 50 cm in length) manually inserted into the soil to a depth of 20 cm. The piston corer used the same methodology as the manual push corer with the assistance of a vacuum piston and an extendable handle (up to 5 m) to sample underwater sediments in the river channel or marsh soils in greater than 1 m of water. Samples were extruded in the field in two sections (0–10 cm and 10–20 cm increments). For the purposes of this study, only data from the 0–10 cm increment were used. For the purposes of this study, soils were analyzed for bulk density, loss-on-ignition, TP, TN, and total carbon (TC). Soil samples were dried at 70 °C and weighed to determine the moisture content and bulk density. Total P content was determined from digestate concentrations prepared by ashing and 1.0 M HCl hot-block digestion using the ascorbic acid automated colorimetric procedure (Method 365.1; [27]). Total carbon and nitrogen were determined on oven-dried, ground samples using a Flash EA 1112 Elemental Analyzer (CE Instruments, Upper Saddlebrook, NJ, USA). Loss-on-ignition (LOI) was calculated using percent ash values subtracted from 100%.

*2.3. Data Analysis*

To evaluate long-term changes in nutrient concentrations and loadings, annual total discharge volumes, TP and TN loads, and flow-weighted mean (FWM) concentrations were calculated for each water control structure along the Kissimmee River. Loads and FWM concentrations were estimated by interpolating nutrient concentrations daily from grab samples collected at each respective structure during days of observed discharge. Daily interpolated nutrient concentrations were then multiplied by daily flow and summed for each WY. Annual FWM values were calculated by dividing the total annual nutrient load by the total annual flow. Mann–Kendall trend analysis was performed on annual TP and TN FWM concentrations and loads for each structure using the '*kendallTrendTest*' in the EnvStats R-package [28].

The nutrient load export of the Kissimmee River floodplain was estimated as the difference of TP and TN loads between S65E and S65 (i.e., S65E–S65). To determine changes in the load export of the floodplain, segmented regression was applied to the cumulative annual nutrient export using the '*segmented*' function in the segmented R-package [29].

To evaluate the spatial distribution of surface sediment (0–10 cm) TP, TN, and TC within the Kissimmee River floodplain, specifically pools C and D (i.e., restoration Phases I and II/III), geostatistical methods were used. Variogram models were fit using the variogram function in the gstat R-package [30]. The parameters of the variogram model are the nugget, sill, and range, where the nugget describes the small-scale variability in the data, the sill is the maximum variability between point pairs, and the range is the distance after which data are no longer correlated [31]. Models with relatively high goodness of fit ($R^2$) values were used for ordinary kriging. The spatial structure or nugget-to-sill ratio (NSR) for each variogram was evaluated by comparing the nugget to the sill. To improve variogram fit, TP concentrations were log-transformed. The final kriged surface was back-transformed and corrected, consistent with methods used in prior studies [32,33].

To provide a finer-resolution spatial interpolation, residual kriging was also used. Using vegetation and landform spatial data [34] (Figure 1), average soil property values were calculated for each of the unique vegetation and landform classifications. For each property, the average values were paired with existing polygons and converted to a raster file. At each sampling location, the average soil property was subtracted from the measured soil property (i.e., the residual value). These residuals were then modeled using ordinary kriging as identified above. The residual model interpolation was then added to the corresponding spatial average values.

All plots were generated in base R and tables were formatted using the "flextable" R-package [35]. All statistical operations were performed using R (Ver 4.0.4, R Foundation for Statistical Computing, Vienna, Austria). Unless otherwise stated, all statistical operations were performed using the base R library. The critical level of significance was set at a = 0.05.

**3. Results**

*3.1. Water Quality Conditions*

Over the past 44 water years, annual nutrient loads entering the Kissimmee River floodplain via Lake Kissimmee (i.e., S65) range from 834 to 211,233 kg TP $Y^{-1}$ and 26,218 to 3,515,506 kg TN $Y^{-1}$. Watershed area normalized nutrient loads range from 2.0 to 504 kg TP $km^{-2}$ $Y^{-1}$ and 63 to 8387 kg TN $km^{-2}$ $Y^{-1}$ (Table 2). Meanwhile, nutrient loads leaving the KRR floodplain to Lake Okeechobee (i.e., S65E) range from 10,436 to 348,449 kg TP $Y^{-1}$ and 112,253 to 4,125,749 kg TN $Y^{-1}$. Floodplain area normalized nutrient loads range from 6.0 to 201 kg TP $km^{-2}$ $Y^{-1}$ and 64.7 to 2379 kg TN $km^{-2}$ $Y^{-1}$ (Table 2).

**Table 2.** Total phosphorus (TP) and total nitrogen (TN) annual load and flow-weighted mean (FWM) concentration summary statistics for water control structures along the Kissimmee River floodplain. Summary statistics include arithmetic mean (mean), minimum (min), maximum (max), standard deviation (SD), and sample size (*N*).

| Variable | Parameter | Statistic | S65 | S65A | S65B | S65C | S65D | S65E |
|---|---|---|---|---|---|---|---|---|
| Load | TP (kg yr$^{-1}$) | Mean | 56,499 | 62,577 | 53,038 | 74,270 | 118,647 | 138,426 |
| | | Min | 834 | 3980 | 8 | 4040 | 8597 | 10,436 |
| | | Max | 211,233 | 165,613 | 166,524 | 203,444 | 383,061 | 348,449 |
| | | SD | 45,451 | 41,589 | 39,070 | 49,958 | 79,803 | 82,252 |
| | | *N* | 44 | 44 | 23 | 39 | 44 | 44 |
| | TN (kg yr$^{-1}$) | Mean | 1,238,417 | 1,277,163 | 1,289,658 | 1,375,850 | 1,625,018 | 1,685,533 |
| | | Min | 26,218 | 62,151 | 138 | 111,078 | 126,163 | 112,253 |
| | | Max | 3,515,506 | 3,044,856 | 3,488,465 | 3,422,184 | 3,922,393 | 4,125,750 |
| | | SD | 789,083 | 736,539 | 912,158 | 840,922 | 968,248 | 994,486 |
| | | *N* | 44 | 44 | 23 | 39 | 44 | 44 |
| FWM | TP (µg L$^{-1}$) | Mean | 58.0 | 64.5 | 55.7 | 69.5 | 94.6 | 111.9 |
| | | Min | 30.7 | 34.3 | 31.8 | 34.7 | 57.4 | 65.5 |
| | | Max | 94.3 | 134.8 | 92.2 | 162.9 | 201.2 | 228.0 |
| | | SD | 17.6 | 19.8 | 14.7 | 25.9 | 29.9 | 36.2 |
| | | *N* | 44 | 44 | 23 | 39 | 44 | 44 |
| | TN (mg L$^{-1}$) | Mean | 1.35 | 1.32 | 1.35 | 1.29 | 1.31 | 1.32 |
| | | Min | 0.81 | 0.87 | 0.95 | 0.88 | 0.99 | 0.97 |
| | | Max | 2.19 | 1.90 | 1.95 | 1.97 | 1.96 | 2.02 |
| | | SD | 0.26 | 0.21 | 0.30 | 0.26 | 0.23 | 0.22 |
| | | *N* | 44 | 44 | 23 | 39 | 44 | 44 |

During the 44-water-year period of record, annual TP loads significantly increased for most of the water control structures along the floodplain, except S65B presumably due to the shorter period of record and variability in the annual time-series (Table 3 and Figure 2). Additionally, the annual flow-weighted mean TP concentration significantly increased for most locations, except S65D and S65E (Table 2 and Figure 3). Trends in annual TN loads did not significantly change over the period of record for water control structures along the floodplain (Table 3 and Figure 2). Despite a lack of annual loading trends, the annual flow-weighted mean TN concentration significantly declined for S65B and S65C. Both of these structures were removed as part of the restoration activity (Table 3 and Figure 3).

**Table 3.** Total phosphorus (TP) and total nitrogen (TN) annual load and flow-weighted mean concentration Mann–Kendall long-term trend results for water control structures along the Kissimmee River floodplain.

| Variable | Parameter | Statistic | S65 | S65A | S65B | S65C | S65D | S65E |
|---|---|---|---|---|---|---|---|---|
| Load | TP | Kendall τ | 0.28 | 0.29 | 0.12 | 0.30 | 0.26 | 0.16 |
| | | ρ-value | <0.01 | <0.01 | 0.43 | <0.01 | <0.05 | 0.12 |
| | | Thiel Sen Slope [A] | 1058 | 1206 | 1189 | 2027 | 2237 | 1604 |
| | TN | Kendall τ | 0.11 | 0.11 | −0.11 | 0.10 | 0.14 | 0.06 |
| | | ρ-value | 0.32 | 0.32 | 0.46 | 0.36 | 0.20 | 0.55 |
| | | Thiel Sen Slope [A] | 10341 | 9419 | −27066 | 11055 | 14240 | 7913 |
| FWM | TP | Kendall τ | 0.41 | 0.33 | 0.30 | 0.39 | 0.18 | 0.03 |
| | | ρ-value | <0.01 | <0.01 | 0.05 | <0.01 | 0.08 | 0.78 |
| | | Thiel Sen Slope [B] | 0.72 | 0.60 | 0.90 | 0.93 | 0.50 | 0.09 |
| | TN | Kendall τ | −0.10 | −0.17 | −0.63 | −0.22 | −0.19 | −0.19 |
| | | ρ-value | 0.35 | 0.11 | <0.01 | 0.05 | 0.06 | 0.07 |
| | | Thiel Sen Slope [C] | −0.003 | −0.004 | −0.027 1pt | −0.008 | −0.005 | −0.005 |

[A] Units: kg yr$^{-1}$; [B] µg L$^{-1}$ yr$^{-1}$; [C] mg L$^{-1}$ yr$^{-1}$.

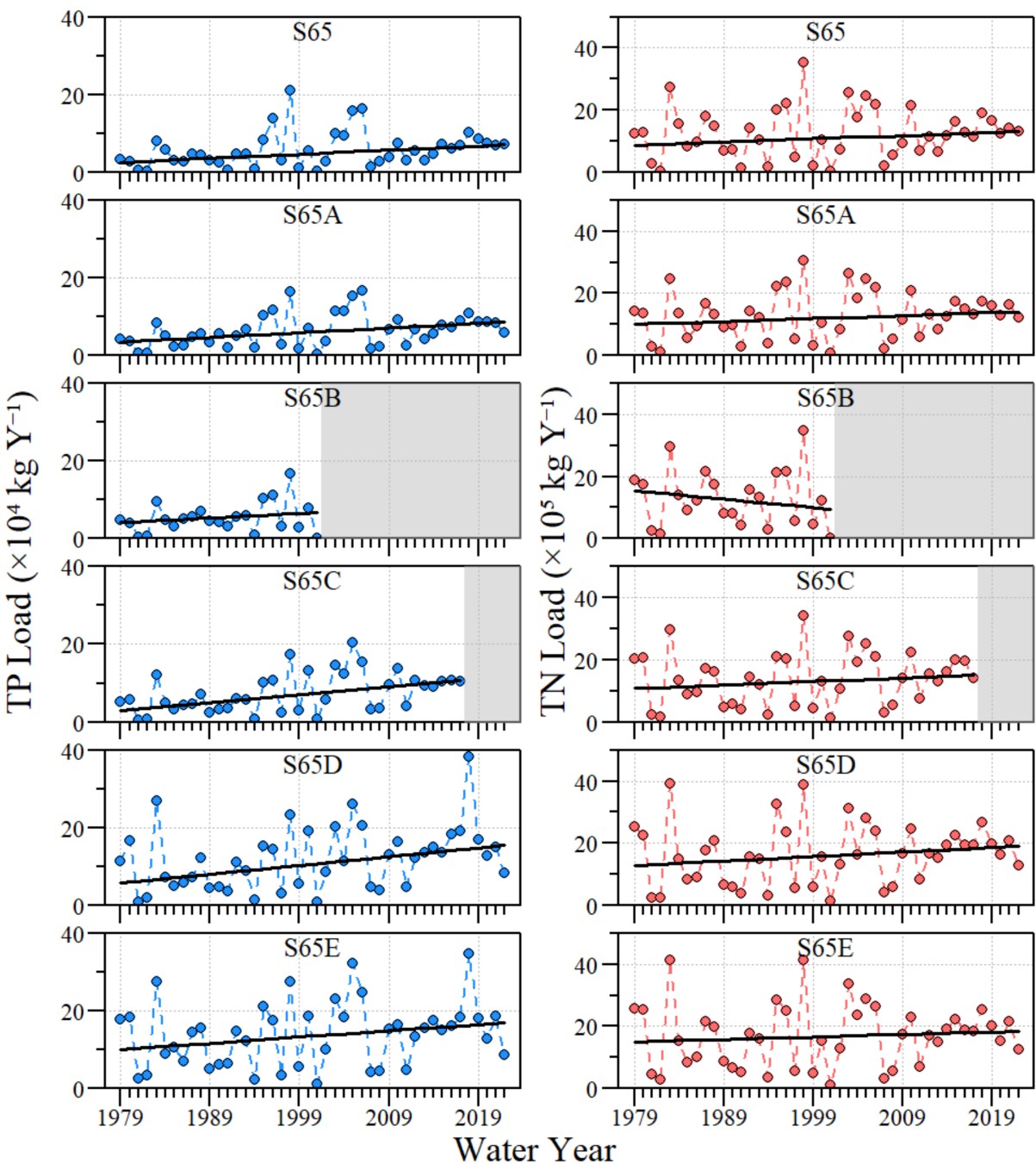

**Figure 2.** Total phosphorus (TP) and total nitrogen (TN) total annual load for each structure along the Kissimmee River floodplain from water year (May–April) 1979 to 2022. The black line through each time-series indicates the monotonic trend in annual values for each respective structure and parameter. Grey-shaded areas for S65B and S65C indicate when the structures were removed as part of restoration efforts.

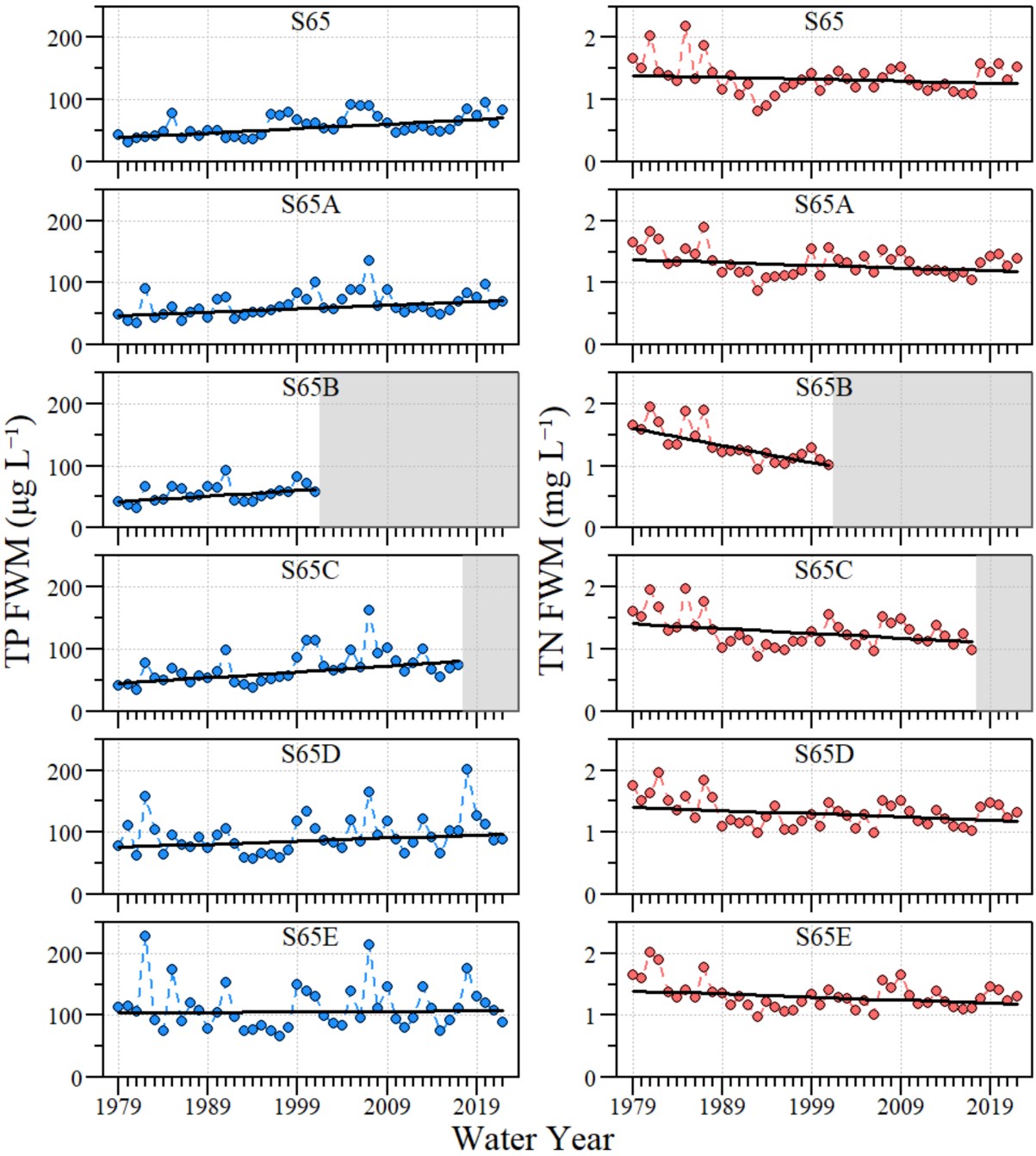

**Figure 3.** Total phosphorus (TP) and total nitrogen (TN) annual flow-weighted mean concentration for each structure along the Kissimmee River floodplain from water year (May–April) 1979 to 2022. The black line through each time-series indicates the monotonic trend in annual values for each respective structure and parameter. Grey-shaded areas for S65B and S65C indicate when the structures were removed as part of restoration efforts.

During the period of record, loads at S65E were always greater than loads estimated for S65 for both TP and TN, suggesting that the Kissimmee River floodplain is a consistent nutrient exporter to downstream water bodies. However, significant breakpoints were detected in the cumulative annual TP export from the floodplain ($R^2 = 1.00$, $F_{(9,34)} = 2869$, $\rho < 0.01$). The breakpoints in the cumulative annual TP export correspond to WY1988, 1999, 2016, and 2018 (Figure 4), corresponding to climatic events (e.g., hurricanes and low/high discharge years) and restoration activities (e.g., backfilling) within the floodplain. Similarly, significant breakpoints were detected in the cumulative annual TN export from

the floodplain ($R^2$ = 1.00, $F_{(9,34)}$ = 2116, $\rho$ < 0.01). The breakpoints in the cumulative annual TN export correspond to WY1983, 1990, 2012, and 2018 (Figure 4). While these breakpoints differ from the TP export breakpoints, they generally correspond to events that affect discharge and loading dynamics.

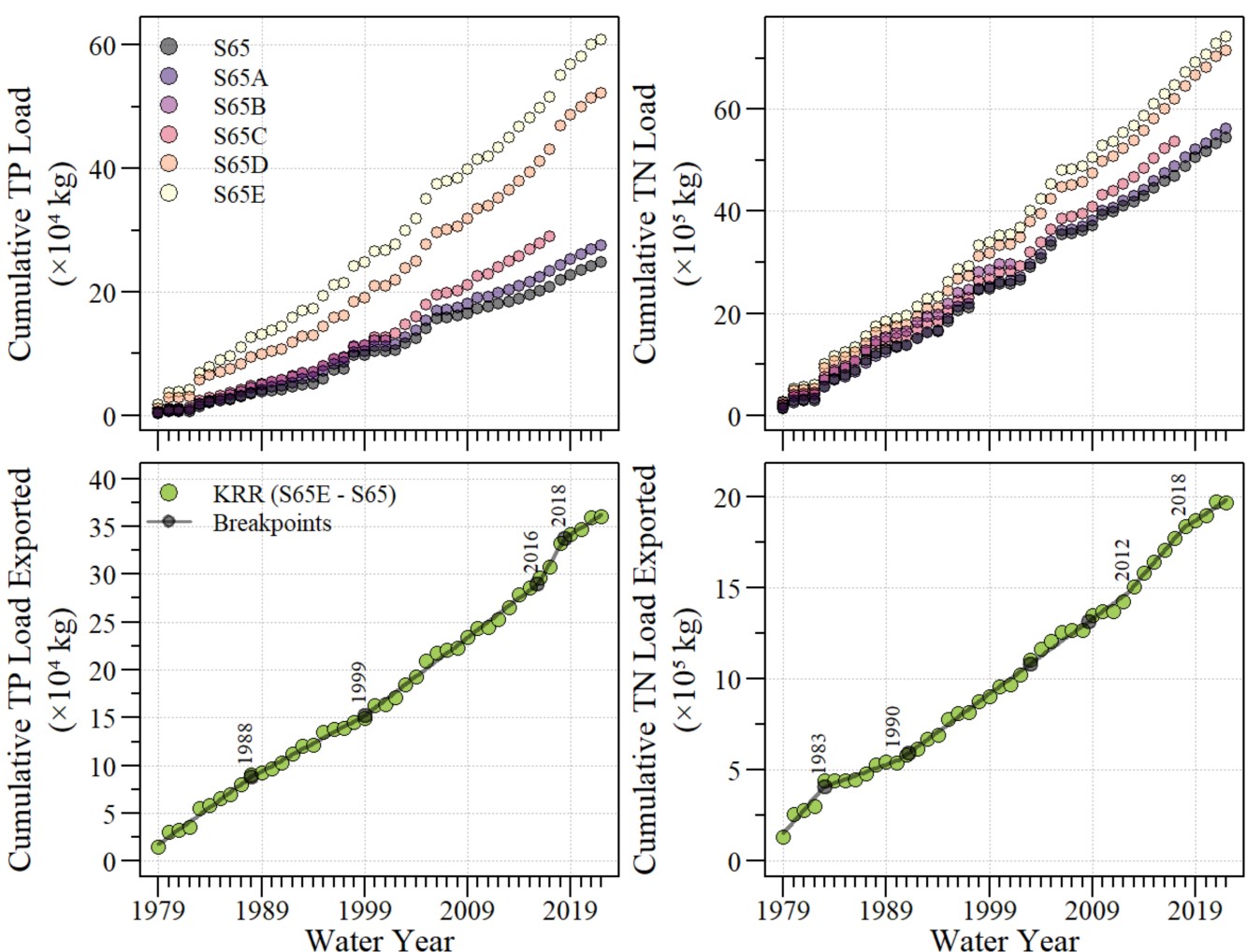

**Figure 4.** Top: cumulative annual load for each water control structure within the Kissimmee River floodplain for total phosphorus (TP, **left**) and total nitrogen (TN, **right**). Bottom: cumulative annual TP and TN load exported from the Kissimmee River during the water year 1979 to 2022 period of record.

## 3.2. Soil Nutrient Distribution

Surface soil (0–10 cm) nutrient concentrations including TP, TN, and TC significantly varied across the floodplain. Sediment TP concentrations ranged from 15 to 3251 mg kg$^{-1}$ with significant differences between vegetation communities ($\chi^2$ = 31.91; df = 5; $\rho$ < 0.01) and landscape form (Figure 5; $\chi^2$ = 1150.10; df = 5; $\rho$ < 0.01). Between the two pools of the floodplain, soil TP concentrations were similar between several vegetative communities with broadleaf marsh having the greatest median concentration and variability (Figure 5). Similarly, soil TP concentrations were similar across the different landscape units with the spoil and backfill landscape units having the greater mean concentrations and variability (Figure 5 and Table 4). Soil TN concentrations ranged from 0.06 to 30.0 g kg$^{-1}$ with significant differences between vegetation communities ($\chi^2$ = 75.8; df = 5; $\rho$ < 0.01) and landscape form (Figure 5; $\chi^2$ = 82.2; df = 5; $\rho$ < 0.01). High soil TN concentrations were observed in aquatic vegetation, broadleaf marsh, and wetland forest vegetation communities and channel, floodplain, and other landscape units (Figure 5 and Table 4). Soil TC concentrations ranged from 0.3 to 473.3 g kg$^{-1}$ and followed similar

patterns to that of TN concentrations (Figure 5 and Table 4) with significant differences between vegetation communities ($\chi^2$ = 64.9 df = 5; $\rho$ < 0.01) and landscape form ($\chi^2$ = 56.6; df = 5; $\rho$ < 0.01).

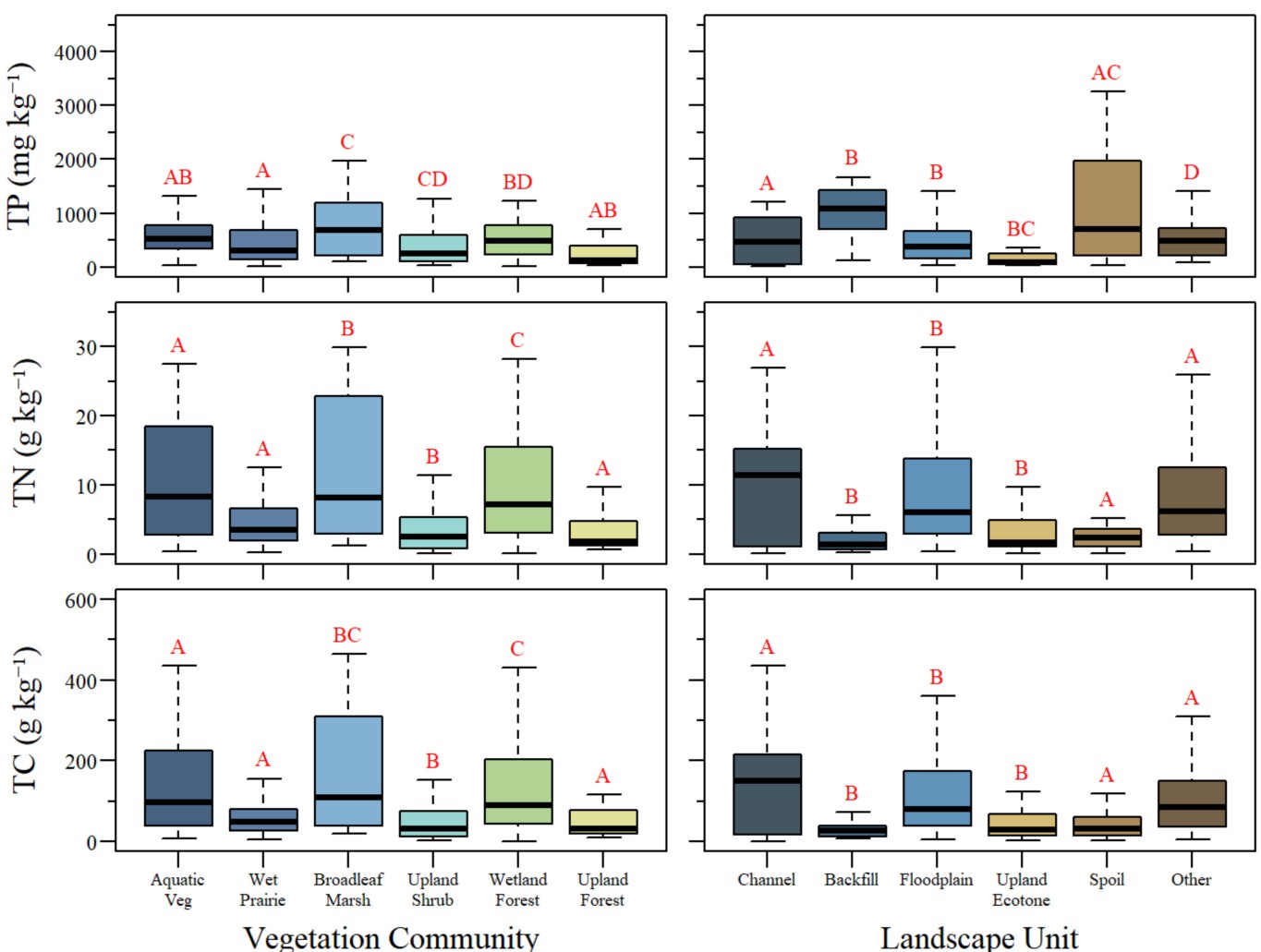

**Figure 5.** Surface sediment total phosphorus (TP, **top**), total nitrogen (TN, **middle**) and total carbon (TC, **bottom**) boxplots grouped by vegetative community (**left**) and landscape unit (**right**). The letters above boxplots indicate statistically significant differences between groups for each parameter. Vegetative community and landscape unit were tested separately.

**Table 4.** Soil summary statistics by vegetation community and landscape unit across the Kissimmee River floodplain (Phases I and II/III). Values are represented as mean ± standard deviation (sample size) for bulk density, loss-on-ignition (LOI), total phosphorus (TP), total nitrogen (TN), and total carbon (TC).

| | | Mean ± SD (*N*) | | | | |
|---|---|---|---|---|---|---|
| **Units** | **Classification** | **Bulk Density (g cm$^{-3}$)** | **LOI (Percent)** | **TP (mg kg$^{-1}$;)** | **TN (g kg$^{-1}$)** | **TC (g kg$^{-1}$)** |
| Vegetation Community | Aquatic Veg | 0.57 ± 0.36 (31) | 29.3 ± 24.0 (35) | 625 ± 448 (35) | 10.7 ± 8.7 (35) | 140.1 ± 120.2 (35) |
| | Wet Prairie | 0.78 ± 0.34 (54) | 17.0 ± 18.5 (58) | 514 ± 558 (58) | 5.9 ± 6.3 (58) | 79.3 ± 93.1 (58) |
| | Broadleaf Marsh | 0.52 ± 0.31 (25) | 33.6 ± 29.0 (29) | 705 ± 531 (29) | 11.7 ± 10.1 (29) | 159.0 ± 141.5 (29) |
| | Upland Shrub | 0.93 ± 0.30 (51) | 11.3 ± 11.3 (58) | 533 ± 710 (58) | 3.7 ± 3.7 (58) | 52.1 ± 56.3 (58) |
| | Wetland Forest/Shrub | 0.52 ± 0.33 (42) | 28.6 ± 25.1 (43) | 591 ± 561 (43) | 9.4 ± 8.1 (43) | 136.2 ± 124.0 (43) |
| | Upland Forest | 0.91 ± 0.28 (28) | 10.4 ± 8.8 (32) | 480 ± 802 (32) | 2.9 ± 2.4 (32) | 49.4 ± 45.3 (32) |
| Landscape Unit | Channel | 0.48 ± 0.31 (17) | 29.0 ± 28.7 (17) | 521 ± 449 (17) | 10.3 ± 9.8 (17) | 142.6 ± 145.3 (17) |
| | Backfill | 0.90 ± 0.34 (13) | 8.4 ± 7.2 (13) | 1003 ± 496 (13) | 2.4 ± 2.5 (13) | 34.9 ± 31.8 (13) |
| | Floodplain | 0.63 ± 0.35 (102) | 26.4 ± 24.4 (113) | 470 ± 379 (113) | 9.1 ± 8.1 (113) | 124.7 ± 119.3 (113) |
| | Upland Ecotone | 0.90 ± 0.30 (38) | 10.3 ± 10.7 (42) | 187 ± 243 (42) | 3.4 ± 3.6 (42) | 51.5 ± 57.4 (42) |
| | Spoil | 0.94 ± 0.28 (34) | 10.3 ± 9.9 (37) | 1183 ± 1089 (37) | 2.9 ± 3.0 (37) | 43.1 ± 42.8 (37) |
| | Other | 0.66 ± 0.37 (27) | 24.1 ± 20.9 (33) | 521 ± 369 (33) | 8.6 ± 7.6 (33) | 116.5 ± 106.4 (33) |

The comparison of soil nutrient concentrations between landscape units (Figure 6) indicates that soils associated with backfill and spoil landscape units contain a higher proportion of TP relative to TC and TN. Overall soil TP is positively correlated with TC (r = 0.65; ρ < 0.01) and TN (r = 0.58; ρ < 0.01). However, these pairwise relationships differ when grouping soils into spoil/backfill and other landscape units. The pairwise relationship between soil TP and TC for spoil/backfill soils (r = 0.49; ρ < 0.01) varies from that of the remaining landscape units (r = 0.88; ρ < 0.01). Similarly, the pairwise relationship of soil TP and TN concentrations for spoil/backfill soils (r = 0.24; ρ < 0.05) differs from that of the other landscape units (r = 0.89; ρ < 0.01). Overall, greater variability in soil TP concentration is apparent for low TN and TC concentrations of spoil/backfill soils. Additionally, high bulk density and low LOI values suggest that the P in these soils is potentially from a geologic origin (deep soils overturned during excavation).

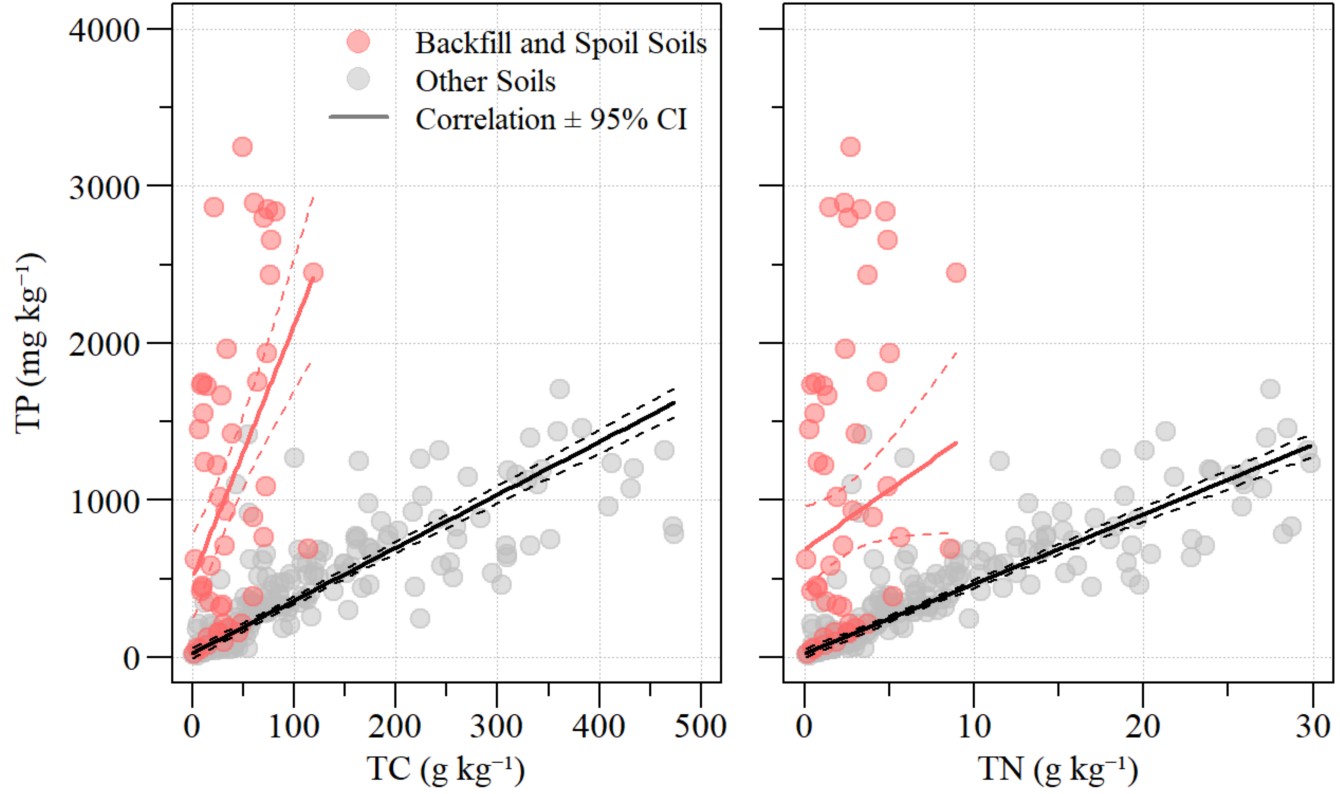

**Figure 6.** Comparison of soil total phosphorous (TP) and total carbon (TC) and TP and total nitrogen (TN) between backfill and spoil soils versus other soils with correlations identified.

### 3.3. Geostatistical Analysis

The spatial autocorrelation of ordinary kriging varied between soil TP, TN, and TC distributions with range values of 2.97, 1.03, and 1.09 km, respectively (Table 4). Meanwhile, the sill-to-nugget ratio for soil TP, TN, and TC was 0.55, 2.22, and 1.85, respectively (Table 5). Based on the sill-to-nugget ratio, soil TP demonstrated a greater proportion of spatial-dependent variation relative to the other soil parameters. Sill-to-nugget ratios of soil TN and TC were greater than 1.0, suggesting a lower proportion of spatial-dependent variation (i.e., a stronger spatial pattern). Visually, this is apparent in the spatial interpolations where ordinary kriging results indicate generally two areas of high soil TP concentrations; meanwhile, TN and TC soil concentrations are relatively variable (Figure 7).

**Table 5.** Semivariogram parameters and kriging prediction error statistics for soil total phosphorus (TP), total nitrogen (TN), and total carbon (TC) using ordinary and residual kriging. Semivariogram model abbreviations: Sph, spherical model; Exp, exponential model.

| Kriging Method | Parameter | Model | Nugget | Sill | Range (m) | Sill:Nugget | RMSE | $R^2$ |
|---|---|---|---|---|---|---|---|---|
| Ordinary | TP | Exp | 0.99 | 0.55 | 2973 | 0.55 | 0.005 | 1.00 |
|  | TN | Sph | 17.8 | 40 | 1035 | 2.22 | 1.8 | 0.96 |
|  | TC | Sph | 4175 | 7733 | 1079 | 1.85 | 23.9 | 1.00 |
| Residual | TP | Sph | 0.43 | 0.18 | 5306 | 0.42 | 0.001 | 1.00 |
|  | TN | Exp | 0.39 | 0.32 | 739 | 0.83 | 0.007 | 0.99 |
|  | TC | Sph | 0.44 | 0.41 | 1292 | 0.93 | 0.030 | 0.93 |

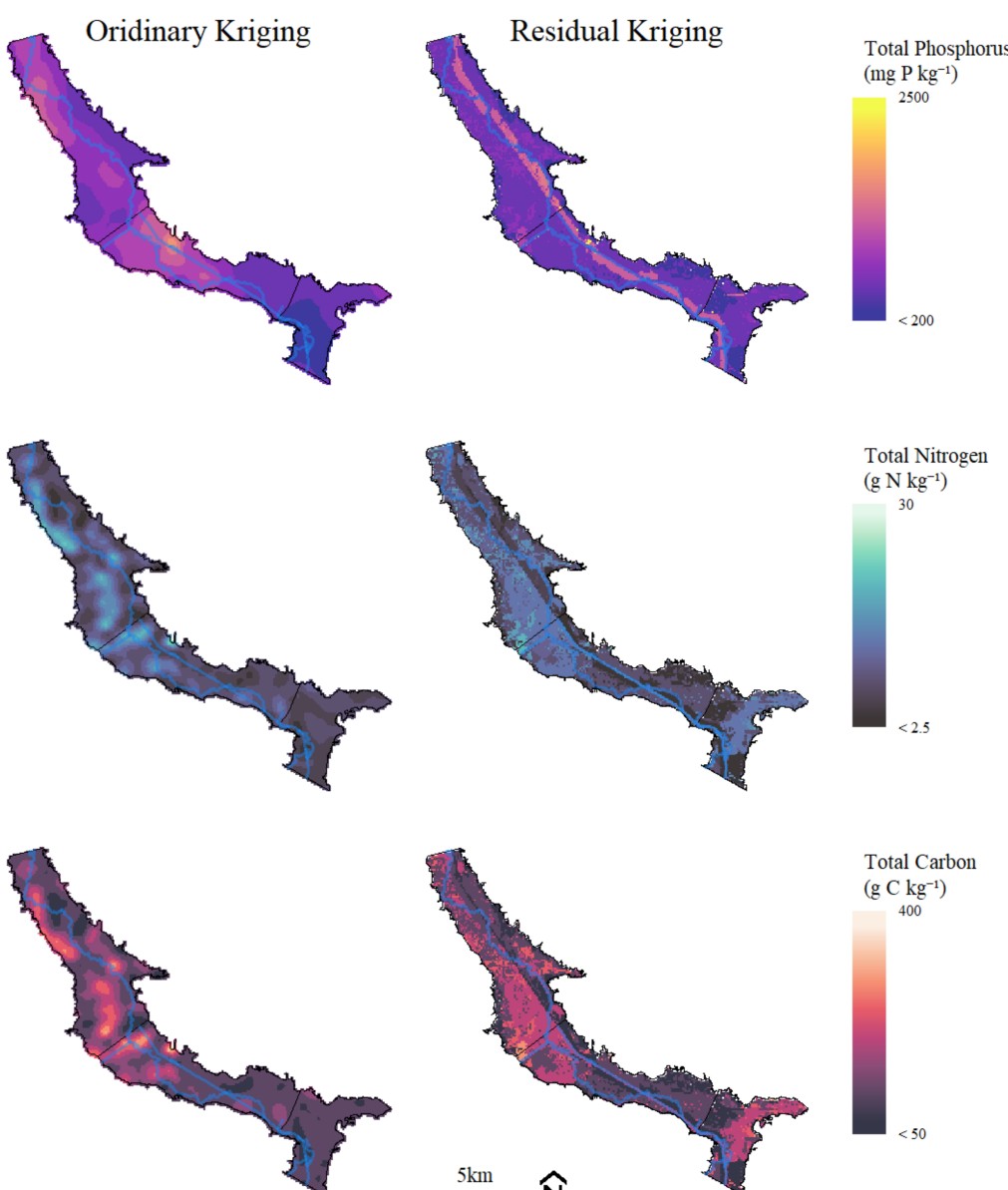

**Figure 7.** Spatial interpolation of soil properties using ordinary (**left**) and residual (**right**) kriging results for total phosphorus (TP, **top**), total nitrogen (TN, **middle**), and total carbon (TC, **bottom**) within the Kissimmee River floodplain.

Spatial patterns of soil TN and TC were very similar but differed from those of soil TP (Figure 7). Extremely high levels of P in spoil material dominate the interpolations, leveraging (masking) those areas over the trends of TP with organic matter observed at the lower concentrations. The overall effect of the highly leveraged TP values in the spoil is a less dynamic and informative mapped surface as the low concentration trends (patterns) are not visible. Given this spatial pattern, residual kriging results provide a finer-scale interpolation of soil characteristics. Moreover, residual kriging further stratifies the landscape to increase the detail and credibility of the interpolations and to reduce the spatial effect of the highly leveraged spoil and backfill materials concerning TP.

As expected, the sill-to-nugget ratios of residual kriging results were relatively low (<1; Table 5). The residual kriging interpolation results of TP indicate very clearly the influence of spoil and backfill materials. However, these areas do not influence the entire landscape as was the case with the ordinary kriging results (Figure 7). The spatial trend observed in the residual kriging of TC and, to some extent, TN suggests higher soil TC and

TN in broadleaf marsh areas of the floodplain wetlands and depressional areas (Figure 7). Meanwhile, the spoil and backfill areas with relatively lower TN and TC values are visible (Figure 7).

## 4. Discussion

Generally, as water moves through a floodplain (i.e., in and out of the main channel) geochemical and microbial processes result in hydrologic and material (e.g., nutrients, biomass) exchanges across the floodplain and to downstream waters. Moreover, the residence time of these exchanges ultimately influences the downstream process and the quality of receiving waters [4]. Modifications to floodplains through channelization or hydrologic manipulation can significantly influence the lateral hydrologic exchange of flow and nutrients, affecting not only the floodplain system but also downstream ecosystems [3,36].

Nutrient mineralization, plant uptake, and sedimentation are important mechanisms of nutrient retention in floodplain ecosystems and are dependent upon floodplain hydro-geomorphology [37,38]. Floodplain ecosystems can trap large proportions of annual river loads by sequestering nutrients, or export less-bioavailable nutrient fractions, from rivers to downstream aquatic ecosystems [30,32]. However, human management and modifications to floodplain ecosystems can reduce hydrologic connectivity and sediment depositional characteristics [39]. In disconnected floodplain ecosystems (e.g., channelized rivers), flood-plain hydrogeomorphology is disrupted by disconnecting geomorphic functional units of varying hydrologic connectivity, potentially reducing the ecological functions [40,41].

In the Kissimmee River floodplain, the system was significantly modified to facilitate water management and navigation. These modifications included channelization of the river and compartmentalization of the floodplain, which almost eliminated flow through the floodplain [42–44]. In doing so, geomorphic functional units along the river were hydrologically disconnected, except in extremely rare circumstances. The disconnected hydrology resulted in a fundamental alteration to plant communities, wildlife habitats, and nutrient cycling. With the combination of physical modifications and water management strategies, nutrients are rapidly transported through the system, resulting in limited in-river–canal interactions. As water moves from Lake Kissimmee, through the Kissimmee River floodplain, and into Lake Okeechobee, nutrient loads and concentrations generally increase (Figures 2 and 4, Table 2). However, notable changes in the loading rates are apparent and could correspond to climate events and/or restoration activity.

### 4.1. River–Floodplain Nutrient Loading

The lake of the Upper Kissimmee Basin (including Lake Kissimmee) has been impacted by point source pollution since the 1950s and 1960s, including four municipal wastewater treatment plants. By the late 1970s, the phosphorus and nitrogen loading to Lake Tohopekaliga was several times greater than natural conditions. Several efforts were initiated to address point-source nutrients in the early 1980s with measurable improvements in water quality in other Upper Bain lakes, including Lake Kissimmee [45]. The improvement in water quality conditions could explain the breakpoints early in the P and N cumulative load time-series (Figure 4), where the loading rate and nutrient export decrease relative to the prior period. Despite these minor improvements, loading and export continued to increase during the 1990s potentially due to upstream ecosystem management (lake drawdowns), restoration activities, and climatic events (e.g., hurricanes, tropical storms, and droughts) (Table 1). However, the breakpoints for both the N and P cumulative load time-series do not exhibit identical times, suggesting that both N and P are more or less sensitive to hydrological restoration or other actions such as climatic conditions and watershed improvements.

In recent years (since 2019), the rate of TP export from the floodplain has changed presumably due to the progression of restoration activities (Table 1 and Figure 4). Phosphorus in the form of legacy inputs or that is organic or geologic in nature could be mobilized as

more of the system is restored. By the late 2010s, most of the canal backfilling had been completed across the Kissimmee River floodplain. As portions of the canal were incrementally backfilled, the potential for more areas of the floodplain to be hydrated became possible. However, hydrologic restoration in the form of adjusted water management is still needed to fully realize the complete restoration effort (i.e., the Headwater Revitalization Schedule) and meet hydroperiod targets for floodplain vegetative communities [12,16,46]. This partial restoration (i.e., physical) may lead to two potential sources of nutrients as water becomes available to hydrate formerly dry portions of the floodplain and potentially mobilize nutrients from the soil to the water column. The majority of the soils used in the backfilling of the canal were sourced from spoil mounds of the original floodplain soil within the Kissimmee River floodplain. These soils have mineral/inorganic characteristics with high P and low N, C, and OM (Table 4, Figures 4–6). Soils of the Lake Okeechobee Basin contain phosphate minerals such as vivianite, apatite, and others [47] and could be a potential source of P to downstream systems. Additionally, legacy and organic nutrients could be mobilized through the rehydration of oxidized soils as areas become hydrated [11,48]. The in situ sources of nutrients combined with agricultural run-off [21] inputs have caused the assimilative capacity of the Kissimmee River floodplain to be exceeded, therefore making the floodplain a consistent net exporter of nutrients (Figure 4). While still a net exporter, the rate of P export had shifted by 2019 to a decreased cumulative rate, potentially signaling restoration success and/or recovery from recent hurricane impacts (i.e., Hurricane Irma). As floodplain function in the Kissimmee River continues to be restored and managed, additional efforts may be needed to address nutrient inputs and internal legacy nutrients.

### 4.2. Floodplain Soils

In a fully functioning floodplain, the ecosystem's hydrologic connectivity should lead to strong spatial gradients of soil nutrients [37,49]. The interaction of flood inundation, flow hydraulics, and sediment during floods facilitates the distribution of sediment nutrients across the floodplain [50]. Despite being virtually disconnected from the river, the Kissimmee River floodplain does exhibit some spatial gradients of sediment nutrients linked to vegetative communities and landscape units (Figures 5 and 7). More specifically, soils in spoil areas generally have the highest P and lowest N and C (Figure 5). While a significant correlation is expected between soil C and P due to observed associations of P with organic matter in many other regional soils [51–53], the C versus P relationship of soils in spoil and backfilled areas is different to those of other landscape units (Figures 5 and 6). Backfill and spoil soils exhibited low C and N with high P concentrations (Figure 6). The low C associated with these high P values potentially suggests Pliocene-aged P-bearing soil minerals [54]. Moreover, the apparent expansion of TP enrichment in the backfill areas of Phase I suggests that disturbances in spoil materials will increase the overall spatial extent of the elevated soil P and contrasts with the relatively small spatial footprint of high TP materials in Phase II/III (Figure 5).

In addition to variable P distributions along the Kissimmee floodplain, soil N and C concentrations were highly variable between vegetation communities and landscape units (Figure 5). Most notably, floodplain and channel landscape units had relatively higher N and C concentrations, suggesting the potential surficial accumulation of organic matter. However, restoration discharges are expected to "flush" organic debris and sediment from the active river channel [11]. The interaction of flood dynamics/seasonal inundation and biological succession creates a dynamic mosaic of biogeochemically active patches across the landscape [55]. As such, the soil N and C distributions were the greatest in riparian and floodplain wetlands (Figure 5). Consistent with past studies [55,56], these results suggest that landscape position is a strong driver in floodplain organic matter and nitrogen dynamics.

## 5. Conclusions

Despite being hydrologically disconnected for more than half a century, the Kissimmee River floodplain is consistent with other floodplain ecosystems as soil N concentrations were strongly associated with OM accumulation and P accumulation was strongly correlated with mineral sediment deposition. Floodplains function as important nutrient sinks along the terrestrial–aquatic continuum. However, legacy nutrient impacts and hydrologic disconnections for flood control measures reduce the ability of the floodplain to function and store nutrients. Historic nutrient loading is reflected in the relatively high soil nutrient concentrations observed across the floodplain, but potential geologic sources of P are also evident in soil and backfilled regions of the floodplain that also can contribute to the downstream transport of P from leaching and flooding. In the case of the Kissimmee River and floodplain, historic nutrient inputs and hydrologic disconnection have resulted in the floodplain being a net exporter of nutrients downstream to Lake Okeechobee. Lake Okeechobee is a severely P-impacted system that is plagued by poor water quality, frequent algal blooms (including Harmful Algal Blooms), a significant reduction in submerged aquatic vegetation community cover, and a dwindling fishery. These impacts can also to a degree cascade to other downstream systems such as the northern Estuaries and Everglades ecosystems. Ultimately, physical restoration is not enough to restore floodplain function. While there may be early signs of improvements, hydrologic restoration is also needed to obtain the benefits of complete restoration.

**Author Contributions:** P.J.II performed data analyses including necessary calculations and statistical analyses and wrote the manuscript. T.Z.O. and R.E. were involved with sediment sample collection, initial analyses and writing of the manuscript. All authors have read and agreed to the published version of the manuscript.

**Funding:** Part of this study (sediment sampling) was funded by the South Florida Water Management District under contract number PO4500060021.

**Data Availability Statement:** Data available upon request.

**Acknowledgments:** We would like to thank the South Florida Water Management District and University of Florida staff for providing field and analytical support for the data used in this study. We would also like to thank the peer reviewers and editor(s) for their efforts and constructive review of this manuscript. This research was conducted on lands and waters that have been cared for by innumerable generations of original peoples of the past, whose memory we honor.

**Conflicts of Interest:** The authors declare no conflict of interest.

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
