# Peer review of "Evaluation of Biogeochemical Changes in Channelized and Restored Portions of a Subtropical Floodplain"

_2673-9917, doi:10.3390/hydrobiology2010001_

Round 1
Reviewer 1 Report
Comments:
The manuscript evaluated the effects of incremental wetland restoration in a river floodplain. Overall, the manuscript is well written with sufficient data covering water and soil in the floodplain. It is also within an interesting topic regarding floodplain nutrient retention evaluation. I would recommend an acceptation after minor revision.
1、 Please show the C38 canal, the flow direction clearly, in Figure. And it is a bit strange to miss Figure 1 below 2.1.
2、 Line 163, Why the loading export is calculated using S65D-S65 rather than S65E?
3、 Figure 4, it seems that after breakpoints, the loading exports generally increased (with larger slope), is it because of the loading or insufficient active floodplain? Could add some discussion.
Author Response
Thank you for your review and input. Below are responses to your comments with the original question included for reference.
1. Please show the C38 canal, the flow direction clearly, in Figure. And it is a bit strange to miss Figure 1 below 2.1.
- The C38 canal and general flow direction have been added to Figure 1 for clarification.
2. Line 163, Why the loading export is calculated using S65D-S65 rather than S65E?
- Originally, the difference between S65D and S65 rather than S65E and S65 was used as the majority of the floodplain and restoration activities reside between S65D and S65. However, in light of this comment and after reviewing the data this analysis has been revised to include S65E rather than S65D. Overall, not much has changed with respect to the breakpoints but the cumulative load exported from the Kissimmee River floodplain area has increased, especially for TP. As such figure 4 and the associated text has been completed.
3. Figure 4. it seems that after breakpoints, the loading exports generally increased (with larger slope), is it because of the loading or insufficient active floodplain? Could add some discussion.
- In some cases especially post-1999 and post-2016 for TP it seems that the cumulative load slope has increased. We believe that this increase is a result of both loading and insufficient active floodplain given when these change points occurred. Additional text will be added.
Reviewer 2 Report
The paper contains long-term analyses of changes in the wetland area. Most of the presented observations are already described in the literature, so I have the impression that the article lacks novelty. I believe that the authors are able to counter this impression by highlighting the main achievements of their study.
In the introduction, the problem should be presented in more detail. At present, the authors have focused on defining the slavic areas and the area of the shaft study.
line 67 here there is no link between the different parts of the text
Figure 1 is of poor quality. Could you please indicate where the C38 canal is located?
Line 157 what does the phrase "water quality concentration" mean?
Line 196 please avoid inserting discussion elements
In table 2 there are abbreviations not explained in the title
Figure 5 please explain the levels of significance coded by each letter
Line 424 what is meant by the phrase "landscape position"
In the conclusions, please avoid quotations, and focus on presenting the most important achievements of the study.
Author Response
Thank you for your review and input. below are response to your comments with the original questions included for your reference.
In the introduction, the problem should be presented in more detail. At present, the authors have focused on defining the slavic areas and the area of the shaft study.
- Noted, the introduction section will be edited to include a finer detail of the problem. However, the authors were unclear with the comment concerning “slavic areas” and “shaft study”.
line 67 here there is no link between the different parts of the text
- Noted, text has been edited to address this concern. Thank you. The sentence “This hydrology and associated hydrodynamics shaped the flora and fauna of the Kissimmee River floodplain.” was added.
Figure 1 is of poor quality. Could you please indicate where the C38 canal is located?
- The C38 canal and general flow direction have been added to Figure 1 for clarification.
Line 157 what does the phrase "water quality concentration" mean?
- “water quality concentration” was edited to “nutrient concentration”
Line 196 please avoid inserting discussion elements
- It is unclear what discussion elements are being referenced as this line and paragraph is reporting statistical results.
In table 2 there are abbreviations not explained in the title
- Definitions for flow-weighted mean, and summary statistics were included. Thank you.
Figure 5 please explain the levels of significance coded by each letter
- Thank you for catching this, text has been added in the methods section regarding the critical level of significance.
Line 424 what is meant by the phrase "landscape position"
- This text was changed to “landscape characteristics and ecological communities” to reflect the landscape unit analysis performed in the soil residual kriging interpolation.
In the conclusions, please avoid quotations, and focus on presenting the most important achievements of the study.
- Noted, the text has been revised.
Round 2
Reviewer 2 Report
Thank you for proofreading the manuscript. I have no further comments.